# Nutrient-induced acidification modulates soil biodiversity-function relationships

Zhengkun Hu[1,2], Manuel Delgado-Baquerizo [3], Nicolas Fanin[4], Xiaoyun Chen[1], Yan Zhou[1], Guozhen Du[5], Feng Hu[1], Lin Jiang [6], Shuijin Hu [7] & Manqiang Liu [2]✉

Nutrient enrichment is a major global change component that often disrupts the relationship between aboveground biodiversity and ecosystem functions by promoting species dominance, altering trophic interactions, and reducing ecosystem stability. Emerging evidence indicates that nutrient enrichment also reduces soil biodiversity and weakens the relationship between belowground biodiversity and ecosystem functions, but the underlying mechanisms remain largely unclear. Here, we explore the effects of nutrient enrichment on soil properties, soil biodiversity, and multiple ecosystem functions through a 13-year field experiment. We show that soil acidification induced by nutrient enrichment, rather than changes in mineral nutrient and carbon (C) availability, is the primary factor negatively affecting the relationship between soil diversity and ecosystem multifunctionality. Nitrogen and phosphorus additions significantly reduce soil pH, diversity of bacteria, fungi and nematodes, as well as an array of ecosystem functions related to C and nutrient cycling. Effects of nutrient enrichment on microbial diversity also have negative consequences at higher trophic levels on the diversity of microbivorous nematodes. These results indicate that nutrient-induced acidification can cascade up its impacts along the soil food webs and influence ecosystem functioning, providing novel insight into the mechanisms through which nutrient enrichment influences soil community and ecosystem properties.

Biodiversity critically regulates diverse biogeochemical and ecological processes that sustain ecosystem productivity and stability[1–3]. In natural ecosystems, co-existing species perform various individual functions, which in turn underpin the multiple goods and services provided to human societies[4–6]. However, anthropogenic disturbances such as nutrient enrichment often reduce species richness[7,8], which in turn may weaken the coupling between biodiversity and ecosystem functioning (BEF)[9–11]. For instance, Hautier et

al[10]. showed that nutrient additions diminished the positive effects of plant diversity on the temporal stability of productivity from 42 grasslands across the globe. However, although considerable efforts have been made to understand how plant communities and ecosystem processes responded to nutrient enrichment[12–14], it is still unclear whether similar trends or patterns of the nutrient effects occur in the belowground subsystem. In particular, how nutrient enrichment may influence ecosystem functioning through its impact on soil biota

[1]College of Resources and Environmental Sciences, Nanjing Agricultural University, Nanjing 210095, China. [2]Centre for Grassland Microbiome, State Key Laboratory of Herbage Improvement and Grassland Agro-Ecosystems, College of Pastoral Agriculture Science and Technology, Lanzhou University, Lanzhou 730020, China. [3]Laboratorio de Biodiversidad y Funcionamiento Ecosistémico. Instituto de Recursos Naturales y Agrobiología de Sevilla (IRNAS), CSIC, Av. Reina Mercedes 10, E-41012 Sevilla, Spain. [4]INRAE, Bordeaux Sciences Agro, UMR 1391 ISPA Villenave-d'Ornon, France. [5]College of Ecology, Lanzhou University, Lanzhou 730000, China. [6]School of Biological Sciences, Georgia Institute of Technology, Atlanta, GA, USA. [7]Department of Entomology & Plant Pathology, North Carolina State University, Raleigh, NC, USA. ✉e-mail: liumq@njau.edu.cn

have been rarely assessed in field, and notably in the context of trophic interactions.

Soil biota is hyper-diverse, containing up to one billion bacterial cells consisting of tens of thousands of taxa, and between dozens and thousands species of fungi, protists, and nematodes in only one gram soil[2,15]. An increasing body of literature has shown that soil biota diversity is essential to maintain an efficient level of ecosystem functioning because of numerous roles played by soil microbes and fauna, such as organic matter decomposition and nutrient cycling[6,16,17]. For instance, Delgado-Baquerizo et al[6]. showed that decreasing soil biota diversity across various ecosystems affected multiple ecosystem functions including plant productivity, nutrient cycling and pathogen control. Yet, unlike different plant species that are typically within a single trophic level, soil organisms live within complex soil food webs, which involve diverse trophic interactions[2,18,19]. Most soil organisms live in/on water films and are particularly sensitive to changes in soil conditions such as pH and nutrient concentrations[20–22]. Furthermore, bacteria and fungi are the dominant consumers of organic matter in soil, and they also function as the core basis of the trophic chains in which nematodes and protists feed on bacterial and fungal biomasses and release nutrients for plants[23]. Yet, the mechanisms through which nutrient enrichment may alter soil biota diversity as well as their trophic interactions have rarely been tested experimentally, and the resulting impact on ecological functions are still relatively unknown.

Soil enrichment of plant essential nutrients (i.e., nitrogen (N) and phosphorus (P)) is among the most profound human-driven environmental changes in terrestrial ecosystems. Globally, human activities through chemical N applications and industrial deposition have at least doubled the reactive N input via natural biological N fixation[24,25]. Also, applications of P fertilizers consistently increase soil P availability and accelerate plant N uptake, thus increasing plant productivity[26,27]. To date, most studies of nutrient enrichment effects on the diversity-function relationships have been plant-oriented and focused on one trophic level[13,17,28]. Nutrient enrichment may affect soil biota diversity and its relationship with ecosystem functioning through three non-mutually exclusive mechanisms (Supplementary Fig. 1). First, high soil available N and/or P often increase plant growth and total carbon (C) input (including both shoot- and root-derived C) belowground[29–31]. As soil microbes are generally C-limited[32], enhanced C availability may stimulate the growth of soil bacteria and fungi, the food bases supporting the complex food webs, and significantly impact diversity-function relationships[33]. Second, an increased availability of some nutrients such as N and P relative to other elements alters nutrient stoichiometry in soil[34] and may favor the competitive dominance of some copiotrophic organisms over others[25,35]. As such, ecosystem functions may be decoupled from soil biota diversity, if soil communities are dominated by only a few abundant species[16,36]. Third, nutrient enrichment, and notably high N input, can cause profound changes in soil physicochemical environments that may critically alter soil biota diversity and plant-microbial interactions. In particular, input of $NH_4^+$ (i.e., the dominant form of fertilizer N inputs) may acidify soils because microbial oxidation of $NH_4^+$ generates proton[37–39]. Accumulation of $H^+$ induces soil acidification, and enhances the solubility of heavy metal (e.g., Al and Mn), which may induce toxicity to microbes and plants[40,41] and suppress microbially-mediated processes[42]. Although these three mechanisms can potentially work individually or in concert to influence the effects of nutrient enrichment on the structure and function of soil biota, we know little about how their relative contribution to ecosystem functioning.

Taking advantage of a long-term fertilization experiment, we tested three hypotheses, each aiming at assess the capacity of one of the three mechanisms by which the effects of nutrient enrichment may impact the diversity and function of soil biota. First, if nutrient enrichment affects microbes mainly through alleviating C limitation to microbes (Mechanism 1 in Supplementary Fig. 1), high levels of N and P additions should increase soil biota diversity, with further enhancements of ecosystem functions. Second, if nutrient (N and/or P) limitation (Mechanism 2 in Supplementary Fig. 1) primarily affects the structure, diversity and activity of soil communities, high levels of N and P additions should decrease the diversity of soil biota, with further repercussions on ecosystem functions such as C and nutrient cycling. Third, if nutrient-induced soil acidification (Mechanism 3 in Supplementary Fig. 1) predominantly regulates the structure, diversity and activity of soil communities, decreased soil pH at higher levels of nutrient additions should reduce the diversity of soil biota, with further repercussions on ecosystem functions.

## Results

Utilizing a long-term (13-yr) experiment of gradient nutrient additions in a Tibetan alpine meadow, we explored how nutrient enrichment affects the diversity-function relationships across multiple trophic levels. A total of 26 parameters were quantified or characterized, including four soil physicochemical properties (labile C, mineral N, available P, and pH) and eight diversity indices of soil biota across multiple trophic levels (bacteria, fungi, and nematodes and their 5 functional guilds). We also measured 14 ecosystem functions directly related to C and nutrient stock (total soil C and N), turnovers of C and nutrient (i.e., the degradation of sugar, chitin, lignin and polymer, and P mineralization), organic matter quality (i.e., using the alkyl:O-alkyl ratio), microbial activity (i.e., soil basal respiration), microbial C and nutrient stocks (microbial biomass C, N and P) and ecosystem stability (i.e., aggregate stability and resistance to plant-parasite nematodes).

Soil labile C content was significantly higher under NP120 (120 g $(NH_4)_2HPO_4$ $m^{-2}$) than under NP30 and NP90 (i.e., 30 and 90 g $(NH_4)_2HPO_4$ $m^{-2}$, respectively), but was not significantly different between the control and nutrient treatments (Fig. 1a). As expected, soil mineral N and available P increased along the NP gradient, from 3.67 to 7.78 mg N·kg$^{-1}$ soil, and from 8.12 to 173.23 mg P·kg$^{-1}$ soil, respectively (Fig. 1b, c). Soil pH declined with increasing NP input from 7.20 in the unfertilized control to 6.54 under NP120 ($P < 0.05$; LSD test; Fig. 1d). The diversity (Shannon diversity index) of all eight soil biota groups (i.e., bacteria, fungi, total nematodes, and bacterivorous, fungivorous, plant parasitic, omnivorous and predatory nematodes) and multi-diversity of soil biota (the average diversity of all soil biota groups) decreased along with the NP gradient (Supplementary Figs. 2 and 3).

Nutrient addition differentially affected ecosystem functions (Supplementary Fig. 4). More specifically, NP addition had no effect on total soil C and total soil N, but significantly increased microbial activity and P-related parameters (microbial biomass P and P mineralization) as P accumulated in soil (Supplementary Fig. 4). NP addition significantly reduced parameters related to C and N cycling (microbial biomass C and N, degradation of sugar, chitin and polymers, the alkyl to O-alkyl ratio) and ecosystem stability (aggregate stability, resistance to parasitic nematode) (Supplementary Fig. 4). In total, 8 out of 14 functions decreased, 3 functions remained unchanged, and the remaining 3 functions increased under high NP input.

To determine the overall impact of nutrient enrichment on diverse ecological functions, we assessed the ecosystem multifunctionality (EMF) through integrating diverse functions into EMF indices[6,43,44]. Utilizing the average approach (see Methods for details), we found that EMF decreased by 11%, 28% and 36% under NP30, NP90 and NP120, respectively (Supplementary Fig. 5a). When EMF was calculated using a multi-threshold approach (see Methods for details), NP120 significantly reduced the number of functions beyond 30%, 50% and 70% thresholds, while NP90 significantly decreased the number of functions beyond the 30% threshold (Supplementary Fig. 5b–d). The number of functions beyond 30%, 50% and 70% thresholds were all positively related to EMF calculated by the average approach ($P < 0.05$; Supplementary Fig. 5e–g), indicating that results from both methods were highly consistent.

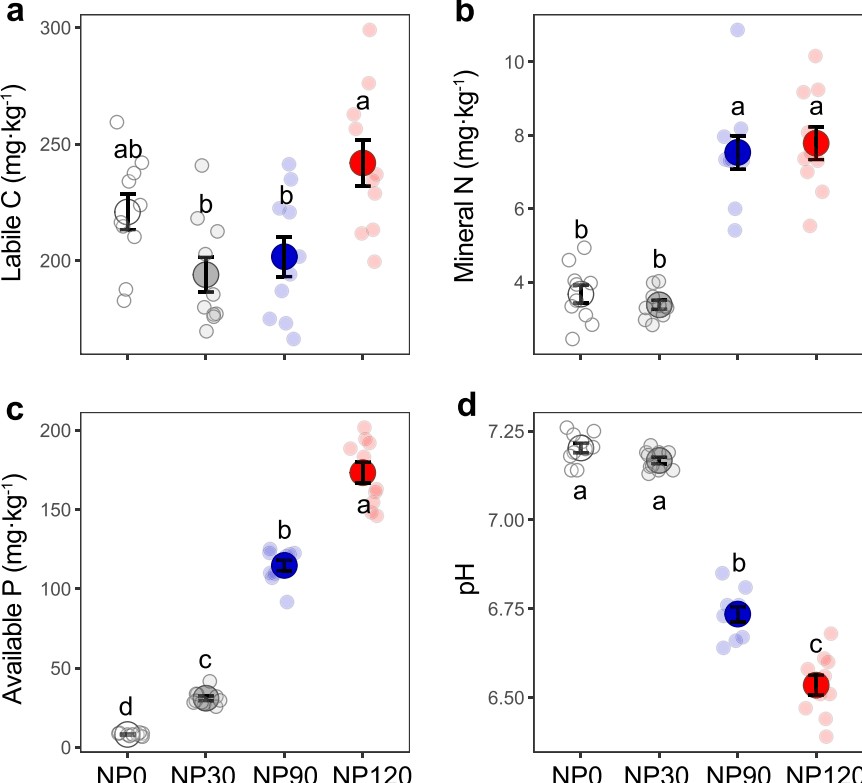

**Fig. 1 | Effects of nutrient enrichment on soil labile C and nutrient content and soil pH.** Difference in (**a**) soil labile C content, (**b**) soil mineral N content, (**c**) soil available P content, and (**d**) soil pH under nutrient enrichment. Dots with bars indicate Mean ± standard error (SE) ($n$ = 10 independent soil samples per treatment), and jittered points represent biologically independent samples for each treatment. Data were analyzed using one-way ANOVA. Based on two-sided tests for multiple comparisons (Fisher's least significant difference), means with different lowercase letters indicate significant difference among treatments ($P < 0.05$). Source data are provided as a Source Data file.

To identify the potential linkages among nutrient-induced changes in physiochemical changes and soil biota, we examined the correlative relationships between soil physiochemical parameters and the diversity of soil organisms. Soil labile C was negatively related to the diversity of soil omnivores ($P < 0.05$; Supplementary Fig. 6), but not to the diversity of bacteria, fungi, total nematode, bacterivores, fungivores, plant parasites, predators (Supplementary Fig. 6), or the multidiversity of soil biota (Fig. 2a). Soil mineral N and available P were negatively related to the diversity of all eight groups of soil biota (Supplementary Figs. 7 and 8), and the multidiversity (Fig. 2b, c). In contrast, soil pH was positively correlated with the diversity of all eight groups of soil biota (Supplementary Fig. 9) and the multidiversity (Fig. 2d).

To examine whether changes in microbial diversity cascaded up along the trophic chains, we examined the relationships between microbes (bacteria and fungi) and microbivores (bacterivores and fungivores), and prey (microbes, bacterivores, fungivores and plant parasites) and predators (omnivores and predators). Our results showed that the diversity of microbes and preys were positively related to the diversity of microbivorous and predatory nematodes, respectively ($P < 0.05$; Fig. 3a, b).

We further explored how nutrient-indued alterations in soil labile C, nutrients and soil pH were related to ecosystem functions. Soil labile C was positively related to P mineralization, microbial activity and aggregate stability, but negatively related to degradation of chitin, lignin and polymer, and resistance to plant-parasites (Supplementary Fig. 10). However, soil labile C was not significantly related to EMF (Fig. 4a). In comparison, soil mineral N and available P were positively associated with P mineralization, microbial biomass P and microbial activity, but negatively related to microbial biomass C and N,

degradation of sugar, chitin and polymers, the alkyl to O-alkyl ratio, and the resistance to plant-parasites (Supplementary Figs. 11 and 12), resulting in overall negative relationships between soil nutrients and EMF (Fig. 4b, c). On the contrary, soil pH was positively related to EMF (Fig. 4d). As to individual functions, soil pH was positively related to microbial biomass C and N, degradation of sugar, chitin and polymers, the alkyl to O-alkyl ratio, aggregate stability, and the resistance to plant-parasites. However, it was negatively correlated with P mineralization, microbial biomass P, and microbial activity (Supplementary Fig. 13). Together, these results showed that increased C and nutrient (N and P) availability under nutrient enrichment did not positively impact most ecosystem functions.

To disentangle the impact of nutrient enrichment on diversity-function relationships, we first assessed the relationships between soil biota diversity and ecosystem functions. For individual functions, the diversity of bacteria, fungi and nematodes was consistently and positively correlated with 9 out of 14 functions: microbial biomass C and N, degradation rates of sugars, chitin, lignin, and polymers, the alkyl to O-alkyl ratio, aggregate stability and parasitic nematode resistance (Fig. 5a). However, the diversity of all soil biota groups was negatively related to three functions that were increased under nutrient addition: microbial activity, microbial biomass P and P mineralization (Fig. 5a). There were significant and positive relationships between the average EMF and the diversity of bacteria, fungi, nematodes, and the whole soil biota, with the multidiversity–EMF relationship explaining more variance than any individual group of soil organisms (Fig. 5b). The significant relationships between soil multidiversity and EMF remained when EMF was calculated by the threshold approach at 30%, 50% and 70% levels (Fig. 5c). We also quantified the influence of nutrient enrichment on

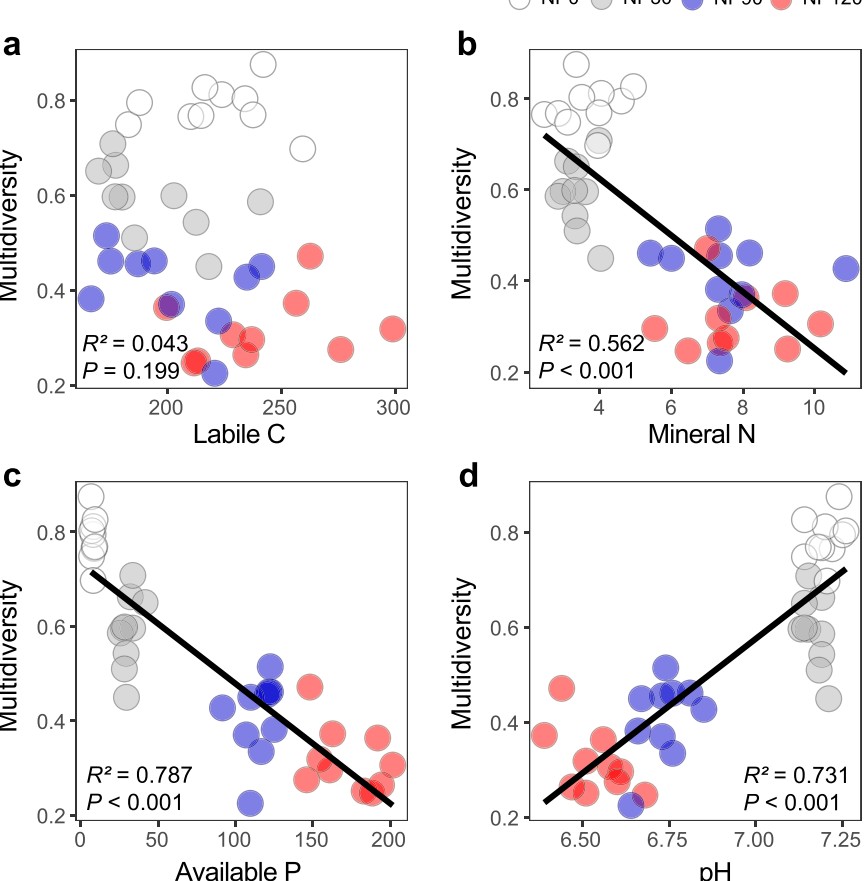

**Fig. 2 | Relationships between soil physicochemical properties and soil multidiversity as influenced by nutrient enrichment. a** Relationship between soil labile C content and multidiversity. **b** Relationship between soil mineral N content and multidiversity. **c** Relationship between soil available P content and multidiversity. **d** Relationship between soil pH and multidiversity. Linear regression model with two-sided test was used for the statistical analysis, and multiple R-squared was used. Relationships are denoted with solid lines and fit statistics ($R^2$ and $P$ values). The solid line represents the significant linear regression ($P < 0.05$). $n = 40$ independent soil samples for regression model. Source data are provided as a Source Data file.

soil diversity-function relationship under the control (NP0), and each of the three nutrient enrichment treatments. In the NP0 control, there were significantly positive relationships between multifunctionality (EMF) and the multidiversity (Fig. 6a) or the diversity of soil bacteria, fungi and nematode (Supplementary Fig. 14a). However, no similar positive relationships between soil biota diversity and EMF were observed in any of the nutrient addition treatments (Fig. 6b–d and Supplementary Fig. 14b–d), suggesting that nutrient enrichment weakened the diversity-function relationships in soil.

We then used the piecewise structural equation modeling (SEM) analysis to assess the relative strengths of direct and indirect relationships among soil labile C, nutrients, pH, soil biota diversity and ecosystem functions. Although NP addition strongly affected soil pH, labile C and soil nutrients, SEM results showed that soil pH predominantly affected EMF through its direct effect on soil biota diversity (pH → microbe, covariance coefficient = 0.84) and nematodes (pH → nematode, covariance coefficient = 0.52) (Fig. 7a). Even more surprising was that mineral N did not directly and significantly affect microbial or nematode diversity. In addition, soil pH indirectly affected EMF through cascading its effects on microbial diversity up to nematode diversity (microbe → nematode, covariance coefficient = 0.34). Both microbial (covariance coefficient = 0.58) and nematode diversity (covariance coefficient = 0.33) positively affected EMF. By calculating standardized total effects of all variables on EMF, we found that soil pH had the largest positive and integrated effects on EMF, followed by microbial and nematode diversity (Fig. 7b).

## Discussion

Results from our long-term field experiment showed that while NP input increased N and P availability (Fig. 1b, c), it reduced soil pH (Fig. 1d) and diversity of soil organisms across multiple trophic levels (microbes, microbivorous and predaceous nematodes) (Fig. 2 and Supplementary Fig. 2), compromising a range of ecosystem functions related to C and nutrient cycling and ecosystem stability (Fig. 4 and Supplementary Fig. 4). Furthermore, nutrient enrichment weakened the positive relationships between soil biota diversity and ecosystem functions (Fig. 6 and Supplementary Fig. 14). Most importantly, we found that soil pH, not labile C and nutrient availability, was a primary driver affecting soil biota diversity and diversity-function relationship, and that soil pH effects cascaded up along the trophic level to influence ecosystem functions (Fig. 7).

Contrary to our first hypothesis, we found no evidence that changes in C availability resulting from nutrient enrichment affected soil biota diversity. Long-term nutrient enrichment often increases soil labile C through enhancing plant growth, litter fall and root exudations[29–31], and stimulate soil microbes, particularly in systems with low soil C. However, soil organic C in our alpine meadow was very high (Supplementary Fig. 4) and low temperature is likely the primary factor constraining microbes and their activities[45,46]. In partial agreement with our second hypothesis and results from other studies showing that nutrient enrichment tends to decrease the diversity of soil biota in grasslands[47,48], we found a negative relationship between nutrient availability and soil biota diversity (Fig. 2 and Supplementary

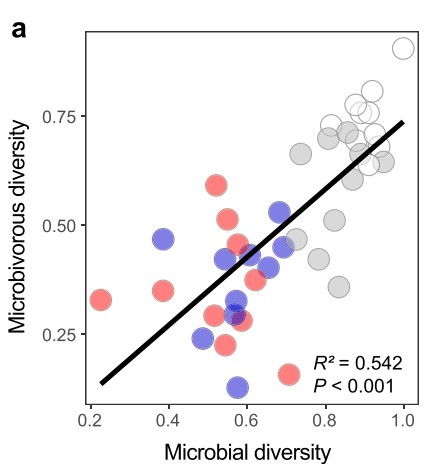
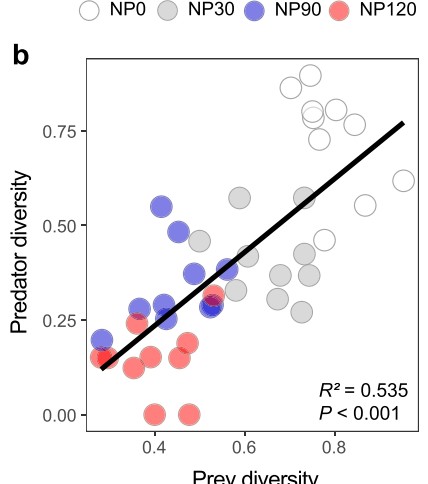

**Fig. 3 | Relationships between soil prey and predator diversity as influenced by nutrient enrichment. a** Relationship between microbial diversity and microbivorous diversity. **b** Relationship between prey diversity and predator diversity. The averaging approach (see "Methods" for details) was also used to calculate microbial and microbivorous nematode diversity, and prey (microbes, bacterivores, fungivores and plant parasites) and predator (omnivores and predators) diversity. Linear regression model with two-sided test was used for the statistical analysis, and multiple R-squared was used. Relationships are denoted with solid lines and fit statistics ($R^2$ and $P$ values). The solid line represents the significant linear regression ($P < 0.05$). $n = 40$ independent soil samples for regression model. Source data are provided as a Source Data file.

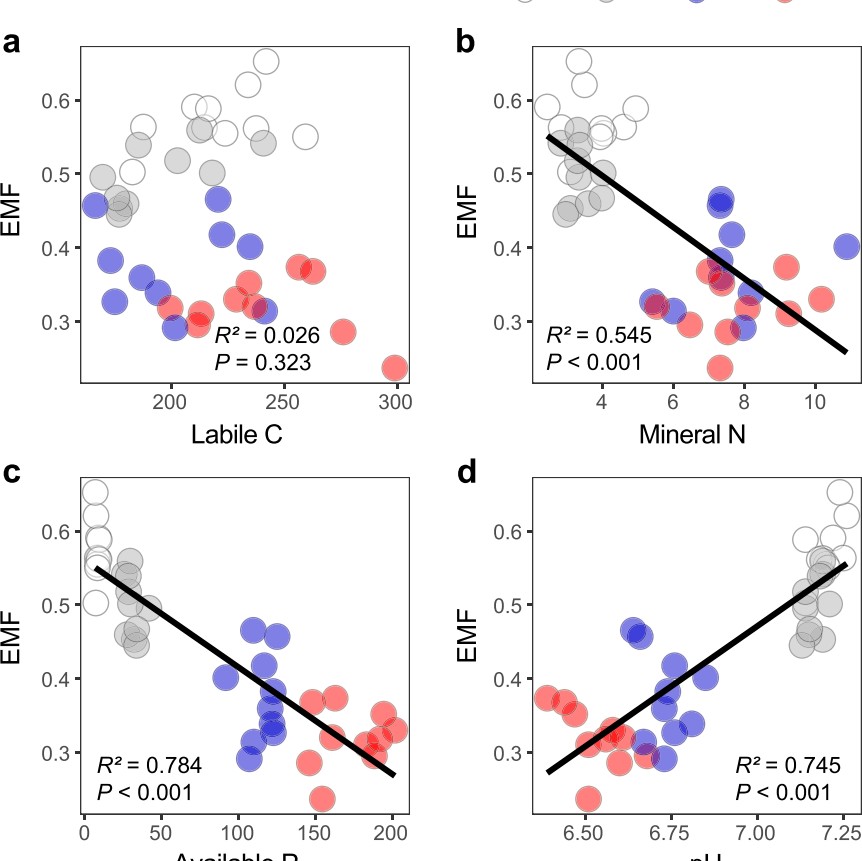

**Fig. 4 | Relationships between soil physicochemical properties and soil ecosystem multifunctionality (EMF) as influenced by nutrient enrichment. a** Relationship between soil labile C content and EMF. **b** Relationship between soil mineral N content and EMF. **c** Relationship between soil available P content and EMF. **d** Relationship between soil pH and EMF. Linear regression model with two-sided test was used for the statistical analysis, and multiple R-squared was used. Relationships are denoted with solid lines and fit statistics ($R^2$ and $P$ values). The solid line represents the significant linear regression ($P < 0.05$). $n = 40$ independent soil samples for regression model. Source data are provided as a Source Data file.

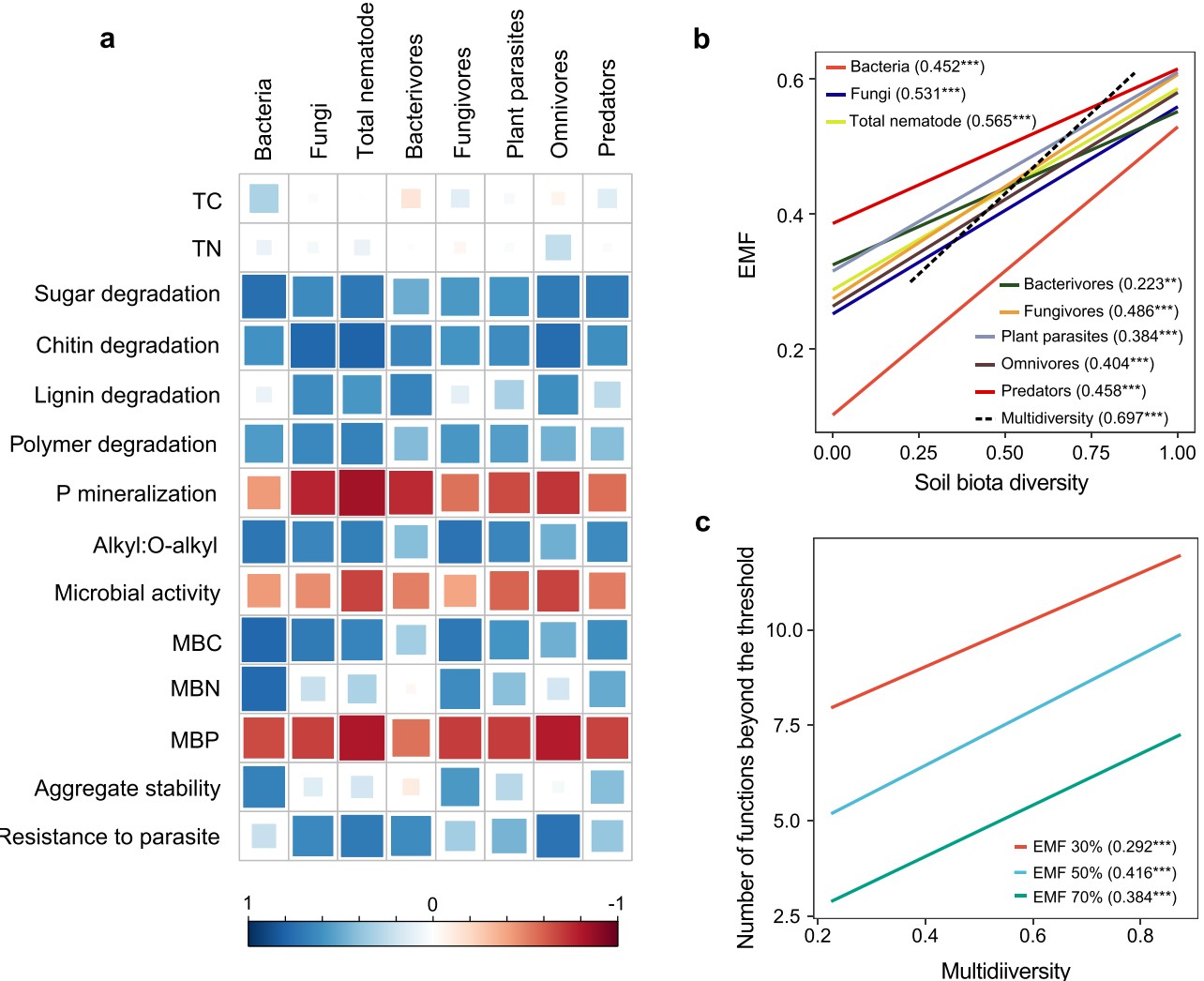

**Fig. 5 | Relationships between soil biodiversity and soil functions or ecosystem multifunctionality as influenced by nutrient enrichment. a** Correlations between the diversity of single groups of soil biota and single ecosystem functions. The heatmap shows significant correlations (calculated by two-sided Spearman's correlation, $P < 0.05$). Color of the circle indicates a positive (blue) or negative (red) correlation, and color intensity indicates the strength of the correlation. **b** The fitted linear relationships between average multifunctionality and biodiversity of selected groups of soil biota and multidiversity. **c** The fitted linear relationships

between multidiversity and the number of functions beyond the threshold of 30% (EMF 30%), 50% (EMF 50%) and 70% (EMF 70%). In (**b**, **c**), linear regression model with two-sided test was used for the statistical analysis, and multiple R-squared was used. Numbers in the parentheses are $R^2$ for the regression and significance levels of each predictor are *$P < 0.05$, **$P < 0.01$, ***$P < 0.001$. $n = 40$ independent soil samples for each regression model. TC total soil carbon, TN total soil nitrogen, MBC microbial biomass carbon, MBN microbial biomass nitrogen, MBP microbial biomass phosphorus. Source data are provided as a Source Data file.

Figs. 7 and 8). Although it has been proposed that nutrient availability could alter the dominance of efficient and competitive microbial species[25,49], we found that the impact of pH overrode nutrient effect when considering all these variables in the same models (Fig. 7). These results suggest that the effect of nutrient availability on soil microorganisms was largely indirect, and at best played only a secondary role in structuring soil communities, highlighting the importance of soil pH in influencing soil biota diversity.

N fertilizers, often in the form of $NH_4^+$-N, induce soil acidification because ammonia oxidizing microorganisms produce $H^+$, while converting $NH_4^+$ into $NO_3^-$ (ref. 37–39). $H^+$ accumulates in soil when $NO_3^-$ leaches out with other cations[50,51]. Furthermore, P fertilizers may also generate $H^+$ in soils with a pH >7.2, which reinforces soil acidification[52]. As such, an increase in soil acidity (i.e., low pH) may increase soluble $Al^{3+}$ (and/or that of other oligo-elements such as Mn), which have long been known to be toxic to bacteria, fungi and plants[40,41,53,54]. What is really surprising is that diversity of all soil biota groups in our study significantly decreased at pH 6.5 under the NP120 treatment. This pH

value is near neutral and would be ideal for many microbes[55,56]. It is possible that the soil microbial community has adapted to the local alkaline environment and decreases in soil pH could induce shifts in the dominant species and community composition[57,58], which are less effective at performing C and nutrient cycling. Together, these results indicate that N-induced acidification may generate shifts in community composition while reducing diversity of soil biota across a range of initial pH levels.

Another major finding of our study is that NP addition reduced soil biota diversity across multiple trophic levels (Fig. 2 and Supplementary Fig. 2). Nutrient enrichment can influence organisms at higher trophic levels through direct and indirect pathways. For instance, nematodes can regulate their osmotic pressure by exchanging ions through the cuticle[59,60]. N-induced soil acidification has likely increased the concentration of ions in soil pore water, which in turn may constrain the ability of nematodes to adjust water state[59,61]. In particular, increases of both $Al^{3+}$ and $H^+$ concentration have been proposed to have direct detrimental effects on soil nematode, particularly for those

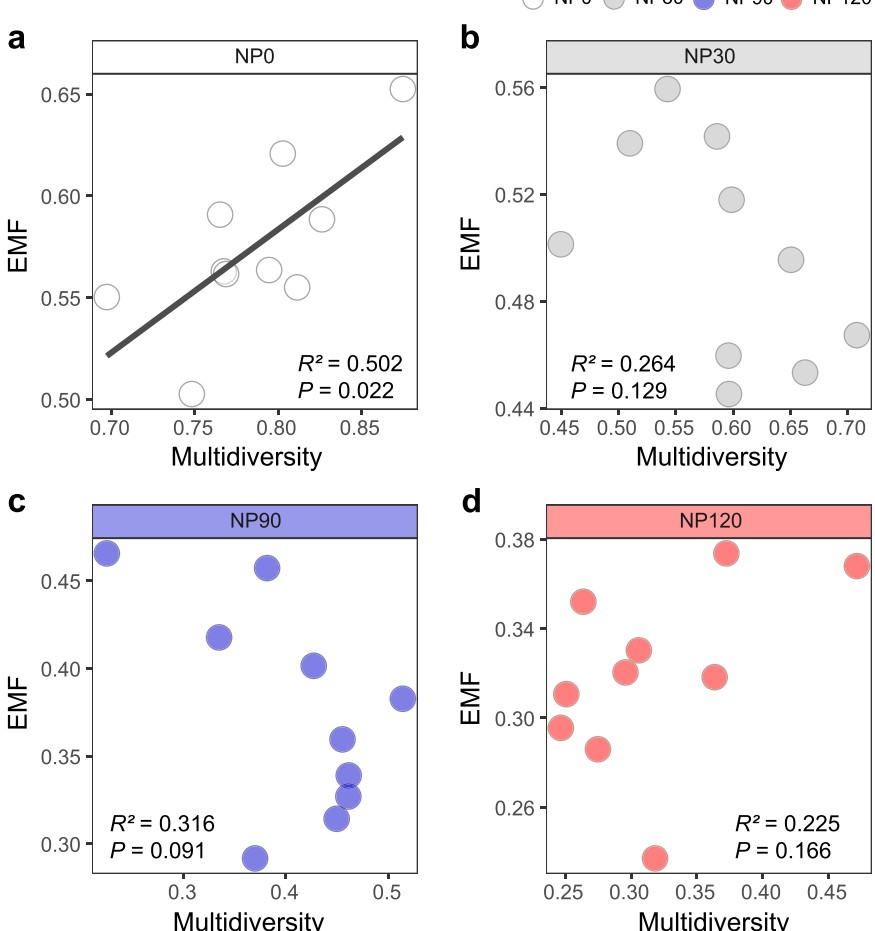

**Fig. 6 | Nutrient enrichment weakened the diversity-function linkage at all three nutrient addition levels.** The fitted linear relationships between soil biodiversity and ecosystem multifunctionality (EMF) under (**a**) NP0, (**b**) NP30, (**c**) NP90, (**d**) NP120. Linear regression model with two-sided test was used for the statistical analysis, and multiple R-squared was used. Relationships are denoted with solid lines and fit statistics ($R$)[2)] and $P$) values). The solid line represents the significant linear regression ($P < 0.05$). $n = 10$ independent soil samples for regression model. Source data are provided as a Source Data file.

at high trophic levels, such as omnivorous and predaceous nematodes[62,63]. Nutrient enrichment can also indirectly affect the diversity and the relative composition of different nematode guilds through cascading up its effects on plants[61,64], and bacteria and fungi[65] (Fig. 3). These changes in diversity of soil biota across the multitrophic levels may have significant implications for ecosystem functions.

In addition to its effects on soil biota diversity, nutrient enrichment modified ecosystem functions (Supplementary Fig. 4), and the biodiversity-function relationships (Fig. 5). The role of soil microbial diversity in maintaining ecosystem productivity and stability has been well documented. Soil microbes control soil organic matter decomposition and nutrient cycling[6,66] and contribute to ecosystem stability through suppressing plant pathogens[6,67] and facilitating soil aggregation[68]. Although many microbes often have high functional redundancy, a decrease in microbial diversity can compromise some ecological functions, especially some specialized functions such as lignin degradation[69] and pathogen suppression[67]. At the same time, microbe-feeding mesofauna such as nematodes and collembola graze bacteria and fungi to facilitate nutrient mineralization and cycling[70,71] and affect the population size of plant parasites and pathogens[72]. In our study, the relationships between soil biota diversity and multifunctionality became tighter when more trophic levels were incorporated (i.e., multidiversity) (Fig. 4b), suggesting that NP effects on soil biota at different trophic levels may converge to affect ecosystem functioning. In addition, we observed that nutrient enrichment not

only altered the general pattern between diversity and functions across the nutrient addition gradient (Fig. 5) but also negated the positive diversity-function relationship under each nutrient addition level (Fig. 6). Nutrient additions have been shown to weaken the diversity-function relationship in grasslands[9,10], but the underlying mechanisms and the mediating drivers remain poorly understood.

Our SEM further revealed that soil pH, not soil nutrient (N and/or P) or labile C availability, predominantly mediated the relationship between soil biota diversity and ecosystem functions under nutrient enrichment. Several mechanisms can account for the changes in soil diversity-function relationship induced by acidification. Alterations in soil pH may directly impact microbial growth (i.e., biomass), physiologies (e.g., respiration) and extracellular enzyme production (Supplementary Fig. 13). Also, soil acidification may differentially affect components of the soil biota. For example, Gram-positive bacteria possess specific mechanisms (e.g., cell envelope alterations and the production of alkali) that enable them to better adapt to soil acidification than Gram-negative bacteria[73]. Because Gram-positive and Gram-negative bacteria have distinct C use strategies, a shift in the bacteria community composition (Supplementary Fig. 15) may influence organic matter decomposition and nutrient cycling[74,75]. Finally, soil acidification can affect soil biota diversity across multiple trophic levels (Supplementary Fig. 9). High acidity can directly affect freeliving nematodes (i.e., microbivores, omnivores and predators) or indirectly by altering their preys. In contrast, it may less affect plant

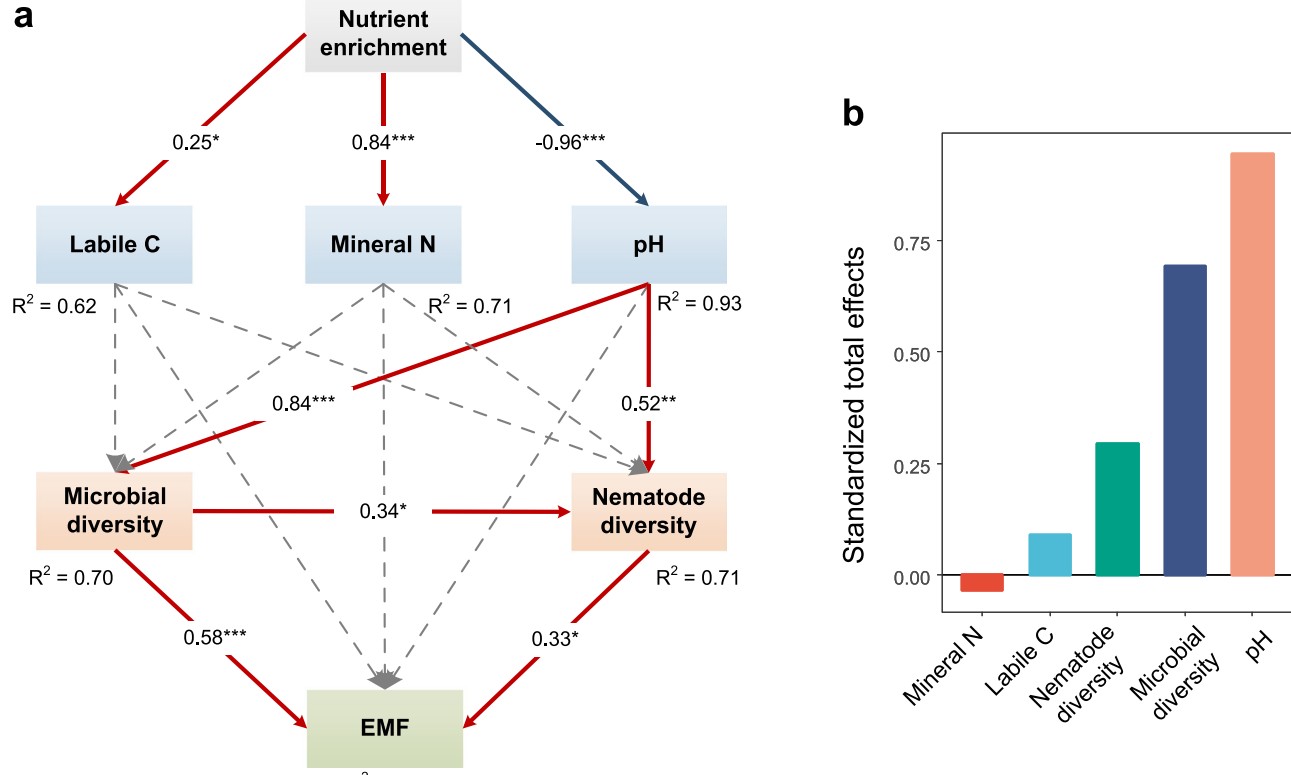

**Fig. 7 | Structural equation modeling describing the direct and indirect effects of nutrient enrichment on ecosystem multifunctionality. a** Structural equation modeling showing the effects of soil abiotic and biotic properties on ecosystem multifunctionality (Fisher's C = 14.87; $P$ = 0.74; df = 20). Statistical significance is based on Fisher's C tests (two-sided) with $n$ = 40 independent soil samples. The red and blue arrows indicate significant positive and negative effects ($P$ < 0.05), respectively, whereas dashed arrows indicate nonsignificant relationships. Values adjacent to arrows represent standardized path coefficients. The width of arrows is proportional to the strength of path coefficients. $R^2$ donates the proportion of variance explained. Significant levels of each predictor are *$P$ < 0.05, **$P$ < 0.01, ***$P$ < 0.001.
**b** Standardized total effects of each individual drivers on ecosystem multifunctionality derived from the SEM depicted above. Source data are provided as a Source Data file.

parasitic nematodes, as suggested by the increased dominance of parasitic nematodes (e.g., *Helicotylenchus* and *Rotylenchus*) under nutrient enrichment (Supplementary Fig. 16). These mechanisms may not be mutually exclusive, but collectively contribute to explaining the observed decrease in EMF in response to nutrient enrichment and the resulting soil acidification.

It should be noted that our experiment site experienced winter grazing by livestock (sheep and yaks). Livestock grazing may alter soil biota diversity and ecosystem functions via aboveground plant tissue removal, dung and urine return, and trampling[76–78]. However, given over 80% of net primary productivity is allocated belowground at our experiment site[79], grazing-induced removal of aboveground plant biomass may have limited effects on soil biota. Also, winter grazing in our field was brief and closely monitored, and dung was manually removed after grazing. In addition, extremely low temperatures (below −10 °C) in winter leads to prolonged freezing of the soil, which mitigated trampling effects on soil. Therefore, winter grazing at our site may not significantly alter the impact of nutrient enrichment on the soil diversity-function relationships. Yet, increasing demand for meat has prompted local nomads to enhance grassland productivity through fertilization in this region, future studies should explore the interactive effects of nutrient enrichment and livestock grazing on soil biota diversity and ecosystem multifunctionality.

Our study presents strong evidence that nutrient-induced changes in soil pH are a primary driver controlling diversity-function relationships. Our results also demonstrate that pH-induced effects cascade up along the trophic chain to affect multiple trophic levels. Given that most previous studies of diversity-function relationships largely focus on one-trophic level, this work advances our understanding of the overall biodiversity effects on ecosystem functioning from a multi-trophic perspective. Furthermore, our results suggest that practices to minimize nitrification to reduce proton generation through synchronizing plant N needs with N supply (e.g., applications of slow-releasing fertilizers or nitrification inhibitors) may alleviate the effects of nutrient inputs on soil biota and sustain soil biota diversity and functions.

## Methods
### Site description
This study was conducted at Walaka (35°58′N, 101°53′E and 3500 m a.s.l.) in the eastern Tibetan Plateau in Maqu County, Gansu Province, China. This alpine ecosystem is characterized by a humid-alpine climate. The mean annual temperature is 1.2 °C, with mean monthly temperatures ranging from −11 °C in January to 11.7 °C in July. Mean annual precipitation is 620 mm. The soil at this site is clay loamy-sand texture (clay 18%, silt 14% and sand 68%). The topsoil layer (0–20 cm) of the meadow contained 36.54 g kg$^{-1}$ organic C, 3.56 g kg$^{-1}$ total N and 4.94 mg kg$^{-1}$ available (Olsen) P and had a pH of 7.64.

### Field experiment design
The experimental site had been fenced since 2001 during growing seasons (May to October) and grazed by sheep and yaks during winters (November to April). The NP addition experiment was established in 2002[80]. There were five levels of the NP addition rate, including a control with no NP addition. Slow-release ammonium phosphate pellets $(NH_4)_2HPO_4$ were applied at the rate of 0 (NP0), 30 (NP30), 60 (NP60), 90 (NP90) and 120 (NP120) g m$^{-2}$ (equivalent to 0, 6.3, 12.6, 18.9, 25.2 g N m$^{-2}$ and 0, 7, 14, 21, 28 g P m$^{-2}$) once a year in May. Twenty-five plots, each measuring 6 m × 10 m, were arranged in a

randomized block design with five replicates of each treatment level. Each plot was separated from the others by a 1-m buffer strip.

## Soil sampling

Soil was sampled from the four treatments in the nutrient enrichment experiment: NP0, NP30, NP90 and NP120. Soil samples were collected in mid-July of 2014 and 2015. Five soil cores (5 cm diameter, 20 cm depth) were taken randomly in each plot, and mixed together to generate one composite sample, resulting in 40 soil samples (4 treatments × 5 replicates × 2 year). Soil samples were immediately transported to the laboratory and then passed through a 2-mm sieve to remove large rocks and roots. All soil analyses requiring fresh material (i.e., nutrient availability and enzyme activities) were done ≤2 weeks after sampling, and all other analyses (i.e., soil C and pH) were done within 2 months after sampling.

## Soil pH, soil dissolved organic C, mineral N and labile P

Soil pH was measured in a mixture comprising a 1:2.5 ratio of soil to deionized water. Labile C was exacted from 10 g fresh soil using 50 mL ultrapure water by centrifugation (7104 × $g$, 10 min). The filtrate that passed through a 0.45 μm filter membrane was analyzed with a total C analyzer (Elementar, Germany). Soil mineral, i.e., $NO_3^-$-N and $NH_4^+$-N, was extracted with 2 M KCl and their concentrations determined using a continuous-flow analyzer (Skalar, Breda, Holland). Available P content was determined colorimetrically using molybdate after extracting samples with 0.5 M NaHCO₃.

## Soil microbes and nematodes

Soil bacterial and fungal diversity were measured by terminal restriction fragment length polymorphism (T-RFLP) analysis[81], which has been used to determine biodiversity in both field and laboratory experiments and has reliable results in diversity estimation[82]. In detail, total soil DNA was extracted from 0.5 g of soil using the FastDNA Spin Kit for Soil and the FastPrep Instrument (MP Biomedicals, Santa Ana, CA, USA), following the manufacturer's instructions. DNA concentration was determined using Nanodrop-2000 spectrophotometer (NanoDrop Technologies Inc. Wilmington, DE, USA). PCR reactions were conducted in duplicate for each DNA sample using the primer pair 27F/1492R and restricted endonucleases *Msp* I for bacterial analysis, and primer pair ITS1F/ITS4 along with *Hha* I for fungal T-RFLP analysis. The fluorescent dye 6-carboxyfluorescein (FAM) was attached at the 5′ end of the 27F and ITS1F primer. Finally, the fragment size was determined at ABI 3130xl Genetic Analyzer (Applied Biosystems). Peaks with fluorescence units <100 were excluded in GeneMapper software (Applied Biosystems) from further analysis due to the basement detection line of the analyzer. Within GeneMapper, the bin width for each fragment was set to 1.5 nt, and the peak area and peak size were used for further statistical analysis.

Soil nematode populations were extracted from 150 g fresh soil using a sequential extraction method[83]. After counting the total numbers of nematodes, 250 specimens were randomly selected from each sample and identified to the genus level base on morphological characteristics (using an Olympus BX50 microscope at 400–1000× magnification). The nematodes were assigned to the following five trophic guilds: bacterivore, fungivore, plant-parasite, omnivore and predator[84]. Nematode genus richness was calculated as the number of genera in each sample. Shannon diversity index ($H'$) that corporate both richness and evenness for bacteria, fungi, and nematodes was calculated:

$$H' = -\sum_{i=1}^{S}(P_i \ln P_i) \qquad (1)$$

where $Pi$ is the proportional abundance of species I, and $S$ is the total number of species.

## Ecosystem functions

Fourteen ecosystem functions that are important to ecosystem productivity and stability were quantified, including two related to C and nutrient stock (total soil C and N), five related to turnovers of C and nutrient (the degradation of sugar, chitin, lignin and polymer, and P mineralization), one related to organic matter quality (the alkyl:O-alkyl ratio), one related to microbial activity (soil basal respiration), three related to microbial C and nutrient stocks (microbial biomass C, N and P) and two related to ecosystem stability (aggregate stability and resistance to plant-parasite nematodes)[4,6]. The rationale for their selection is presented in Supplementary Table 1. Microbial activity was estimated by soil basal respiration via incubation of fresh field soil (10 g dry soil equivalent)[85]. Soil total C and N were measured by using an elemental analyzer (Elementar, Langenselbold, Germany). Microbial biomass C, N and P were estimated using the chloroform fumigation–extraction method, using extraction factors of $K_{EC} = 0.45$, $K_{EN} = 0.45$ and $K_{EP} = 0.4$ (ref. 86). The activities of β-glucosidase (sugar degradation), *N*-acetylglucosaminidase (chitin degradation), phenol oxidase (lignin degradation) and phosphatase (P mineralization) were measured using 2.75 g fresh soil and a microplate fluorometric assay according to previous protocols[87,88]. Polymer degradation was measured with a BIOLOG Microplate®[89]. The chemistry of soil organic matter was characterized by a combination of solid-state cross-polarization magic-angle spinning (CP/MAS) and ¹³C nuclear magnetic resonance (NMR) spectroscopy. The NMR spectrum was divided into different chemical shift regions, and the alkyl to *O*-alkyl ratio was used to provide an index of the decomposition potential of organic materials, with higher ratios indicating a greater decomposition potential[90]. Aggregate distribution was measured by a wet sieving and size-density fractionation approach[91]. In total, three aggregates size classes were obtained: macroaggregate (>250 μm), microaggregate (250–53 μm) and silt and clay fraction (<53 μm). Aggregate stability was represented by mean weight diameter (MWD), calculated with the following formula:

$$MWD = \sum d^*m \qquad (2)$$

where $d$ is the mean diameter of each fraction size and $m$ is the relative fraction mass of aggregates (%). The inversed abundance of plant parasites was obtained via calculating the inverse of total relative abundance of plant-parasite nematodes[6].

## Ecosystem multidiversity and multifunctionality

The plot-estimated Shannon diversity index of each of eight groups of soil organisms (bacteria, fungi, total nematodes and five nematode guilds) was first standardized to 0–1 according to the following formula:

$$STD = \frac{(X - X_{\min})}{(X_{\max} - X_{\min})} \qquad (3)$$

where STD is the standardized variable and $X$, $X_{\min}$ and $X_{\max}$ are the target variable and its minimum and maximum values across all samples, respectively. Then their average was calculated to obtain a multidiversity index[92]. With this approach, the diversity of each soil group contributed equally to the multidiversity index[6]. This approach was also used to calculate microbial and microbivorous nematode diversity, and prey (microbes, bacterivores, fungivores and plant parasites) and predator (omnivores and predators) diversity. We used both the averaging approach and multi-threshold approach to quantify multifunctionality. Each ecosystem function was first standardized to remove the effects of differences in the measurement scale between functions by 0–1 transformation. Then, their average was calculated to obtain an multifunctionality index[6,44]. Moreover, we calculated the number of functions beyond a given threshold (30%, 50% and 70%)

using the multi-threshold approach described in Byrnes et al.[43], following Delgado-Baquerizo et al.[6].

## Statistical analysis

ANOVAs were used to test the effects of nutrient enrichment on soil labile C, nutrient content and pH. The same models were used to test effects of nutrient enrichment on soil biodiversity, ecosystem functions and multifunctionality index. When significant effects were found, post-hoc tests using Fisher's least significant difference were run. Then, linear regressions were conducted between soil abiotic properties (i.e., soil labile C, nutrients and pH) and eight groups of soil organisms (i.e., bacteria, fungi, total nematode, bacterivorous nematode, fungivorous nematode, plant-parasite nematode, omnivorous nematode and predatory nematode) individually or soil multidiversity (standardized average of the diversity of the eight groups of soil organisms). Linear regressions were also used to test the trophic relationships between microbial and nematode diversity, and between prey and predator diversity. The same models were used to explore the relationships between soil abiotic properties (i.e., soil labile C, nutrients and pH) and single functions or multifunctionality. Further, Spearman correlations between the diversity of each of the eight groups of soil organisms and single functions were also performed. Linear regressions were conducted between EMF and eight groups of soil organisms individually or soil multidiversity. Also, linear regressions were conducted between multidiversity and EMF under each nutrient enrichment treatment to assess the direct impacts of nutrient enrichment on soil diversity-function relationships. Finally, piecewise structural equation modeling (SEM)[93] was used to explore the direct and indirect pathways through which soil abiotic (labile C, nutrient availability and pH) and biota diversity influenced EMF under nutrient enrichment (a priori model; Supplementary Fig. 1). The SEM was fit using linear mixed-effects models where block and sampling year were treated as random factors. We initially formulated an a priori model encompassing all hypothesized pathways (Supplementary Fig. 1), and iteratively simplified it by removing non-significant pathways until arriving at the final model. The adequacy of the final model was assessed by Fisher's C statistic in the "*piecewiseSEM*" package[93] in R 4.2.2. The code is available as Supplementary Information (Supplementary Code).

## Data availability

All data that support the findings of this study (including soil physicochemical properties, soil biota diversity, ecosystem functions and multifunctionality indices) are available in the Figshare database (https://doi.org/10.6084/m9.figshare.25460410.v1). Source data are provided with this paper.

## Code availability

R code used for data analysis has been deposited in the Figshare database (https://doi.org/10.6084/m9.figshare.25460410.v1). The code is also available as Supplementary Information (Supplementary Code).

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

## Acknowledgements

This study was supported by the Fundamental Research Funds for the Central Universities (lzujbky-2022-ct04) and Natural Science Foundation of China (42077047, 32301434). Z.H. acknowledges support from the China Postdoctoral Science Foundation (2022M711657) and the Jiangsu Funding Program for Excellent Postdoctoral Talent (2022ZB326). M.D.-B. acknowledges support from TED2021-130908B-C41/AEI/10.13039/501100011033/Unión Europea NextGenerationEU/ PRTR and from the Spanish Ministry of Science and Innovation for the I + D + i project PID2020-115813RA-I00 funded by MCIN/AEI/ 10.13039/501100011033.

## Author contributions

Z.H. and M.L. designed the study. G.D. established the field experiment. X.C., M.L. and Y.Z. performed the laboratory work. Z.H. and X.C. conducted statistical analysis. Z.H. drafted the manuscript with help from M.D.-B., N.F., F.H., L.J., S.H. and M.L. All authors contributed to the article and approved the submitted versions.

## Competing interests

The authors declare no competing interests.
