## [Peer Review File · Nature Communications]

Reviewers' Comments:

Reviewer #1:

Remarks to the Author:

This study examined the effects of nutrient enrichment on soil properties, soil biodiversity, and multiple ecosystem functions, as well as the relationship between diversity and function, using a 13-year experiment in a Tibetan alpine meadow. The results showed that soil acidification, rather than changes in mineral nutrient or carbon availability, was the primary factor negatively affecting the diversity-function relationship. Nutrient additions reduced soil pH, the diversity of bacteria, fungi, and nematodes, as well as a range of ecosystem functions related to carbon and nutrient cycling. The effects of nutrient enrichment on bacteria and fungi also had a detrimental impact on higher trophic levels, such as bacterivorous and fungivorous nematodes. This study provides strong evidence that nutrient-induced changes in soil pH are the primary drivers of diversity-function relationships. Additionally, the results demonstrate that pH-induced effects cascade up the trophic chain, impacting multiple trophic levels. These findings are an interesting discovery for the fields of soil ecology and global change ecology.

Overall, the experiment design is suitable for examining this topic. The logical structure of the paper is commendable, such as the introduction, testing, and discussion of the three hypotheses, as well as the pH-induced effects that cascade up the trophic chain and affect multiple trophic levels. The results indicate that nutrient enrichment-induced acidification can have cascading effects on soil food webs and affect ecosystem functioning, providing new insight into the mechanism by which nutrient enrichment impacts soil community and ecosystem properties. Before the paper can be published, there are a few minor issues that need to be addressed.

1. Title: It might be better to use "Nutrient-induced acidification" or "Nutrient enrichment-induced acidification" than "Nitrogen-induced acidification". Because the additions of nutrients included both N and P, we do not exactly know if it is nitrogen-induced acidification or phosphorus-induced acidification.

2. Line 138: Please describe the treatments of NP30, NP90, and NP120 here; they are the first time appearing in this manuscript.

3. Line 154: The average approach may provide biased estimations of multifunctionality if there are trade-offs among functions in your dataset. For instance, the same multifunctionality level can be obtained if all functions performed at an intermediate level or if some functions performed at a high value while others performed at a low value. Therefore, to ensure that your estimation of multifunctionality was not biased by a single function driving multifunctionality, I suggest that the manuscript could also consider a threshold approach of multifunctionality. The average and threshold indices of multifunctionality might be highly correlated, and both approaches might provide very similar results.

4. Line 305-306: Based on the field experiment design, the experimental site had been fenced since 2001 during growing seasons (May to October) and grazed by sheep and yaks during winters (November to April). This means that the aboveground plant biomass was grazed by sheep and yaks during winters from 2002 to 2015? If I understand correctly, the authors should provide a small section in the Discussion regarding the potential effects of winter grazing on their main conclusions.

5. The SEM should be performed using Piecewise SEM, which enables the SEM to consider the effects of random blocks, although both approaches provide similar results.

Reviewer #2:

Remarks to the Author:

In this study, the authors utilized a long-term fertilization experiment to investigate the impact of nutrient enrichment on soil biodiversity and ecosystem functions. Their findings highlighted that nitrogen-induced acidification diminishes soil biodiversity and has cascading effects throughout soil food webs, thereby influencing ecosystem functionality. While the subject matter is both relevant and significant, and their experiment compellingly demonstrates acidification as the primary mechanism leading to the decline in soil biodiversity and function due to fertilization, there's a notable discrepancy between their results and conclusion regarding the biodiversity-function relationship (BFR). This discrepancy seems central to the article's theme.

The manuscript commendably addresses the effects of nutrient enrichment on:

1. Soil biodiversity
2. Ecosystem functions

However, it falls short in directly analyzing the influence of nutrient enrichment on the relationship between biodiversity and function. For instance, Figure 5 illustrates a positive correlation between diversity and ecosystem multifunctionality (EMF), but it doesn't elucidate if this correlation is amplified, diminished, or even reversed under fertilization conditions. The title and a key result in the abstract emphasize the BFR but lack direct evidentiary support. I see two potential courses of action:

Reframe the Narrative: The authors could pivot their narrative focus from the BFR to separately address biodiversity and ecosystem functions. The primary conclusion, which posits acidification as the predominant mechanism (instead of the other two processes) influencing the decline of soil biodiversity and EMF under nutrient enrichment, is a substantial contribution in its own right. The effects of fertilization on the BFR needn't be emphasized if direct evidence is absent.

Deepen the Analysis: Should the authors wish to maintain their emphasis on the BFR, it would be advisable to quantify the influence of fertilization on the BFR under each NP treatment. This would truly elucidate the specific effects of fertilization on the BFR, such as whether it enhances or weakens the positive BFR. A salient example of this type of analysis is presented in Hautier et al. (2014), where the relationship between richness and the stability of ANPP in fertilized (Figure 1c-d) versus unfertilized communities (Figure 1a-b) is distinctly illustrated.

In summary, I recommend a major revision, either by reorienting the narrative focus or by enhancing the analysis to directly substantiate the effects of fertilization on the BFR.

Responses to the Reviewers' Comments

REVIEWER COMMENTS

Reviewer #1 (Remarks to the Author):

Comment 1: This study examined the effects of nutrient enrichment on soil properties, soil biodiversity, and multiple ecosystem functions, as well as the relationship between diversity and function, using a 13-year experiment in a Tibetan alpine meadow. The results showed that soil acidification, rather than changes in mineral nutrient or carbon availability, was the primary factor negatively affecting the diversity-function relationship. Nutrient additions reduced soil pH, the diversity of bacteria, fungi, and nematodes, as well as a range of ecosystem functions related to carbon and nutrient cycling. The effects of nutrient enrichment on bacteria and fungi also had a detrimental impact on higher trophic levels, such as bacterivorous and fungivorous nematodes. This study provides strong evidence that nutrient-induced changes in soil pH are the primary drivers of diversity-function relationships. Additionally, the results demonstrate that pH-induced effects cascade up the trophic chain, impacting multiple trophic levels. These findings are an interesting discovery for the fields of soil ecology and global change ecology.

Overall, the experiment design is suitable for examining this topic. The logical structure of the paper is commendable, such as the introduction, testing, and discussion of the three hypotheses, as well as the pH-induced effects that cascade up the trophic chain and affect multiple trophic levels. The results indicate that nutrient enrichment-induced acidification can have cascading effects on soil food webs and affect ecosystem functioning, providing new insight into the mechanism by which nutrient enrichment impacts soil community and ecosystem properties. Before the paper can be published, there are a few minor issues that need to be addressed.

Author's reply: We appreciate your positive and constructive comments. In line with these suggestions, we have made revisions to the manuscript, as explained in detail below.

Comment 2: Title: It might be better to use “Nutrient-induced acidification” or “Nutrient enrichment-induced acidification” than “Nitrogen-induced acidification”. Because the additions of nutrients included both N and P, we do not exactly know if it is nitrogen-induced acidification or phosphorus-induced acidification.

Author's reply: The title has been changed to “Nutrient-induced acidification modulates soil biodiversity-function relationships” in the revised manuscript.

Comment 3: Line 138: Please describe the treatments of NP30, NP90, and NP120 here; they

are the first time appearing in this manuscript.

Author's reply: We added more information about the different treatments [Line 174-Line 176]: “Soil labile C content was significantly higher under NP120 (120 g (NH₄)₂HPO₄ m⁻²) than under NP30 and NP90 (i.e., 30 and 90 g (NH₄)₂HPO₄ m⁻², respectively), but was not significantly different between the control and nutrient treatments (Fig. 1a)”.

Comment 4: Line 154: The average approach may provide biased estimations of multifunctionality if there are trade-offs among functions in your dataset. For instance, the same multifunctionality level can be obtained if all functions performed at an intermediate level or if some functions performed at a high value while others performed at a low value. Therefore, to ensure that your estimation of multifunctionality was not biased by a single function driving multifunctionality, I suggest that the manuscript could also consider a threshold approach of multifunctionality. The average and threshold indices of multifunctionality might be highly correlated, and both approaches might provide very similar results.

Author's reply: Great comment. We have now adopted a multi-threshold approach to assess the impact of nutrient enrichment on multifunctionality (EMF) in the revised manuscript. The multi-threshold approach (Byrnes *et al.* 2014) involves counting the number of ecosystem functions that exceed multiple critical thresholds (30%, 50% and 70% of the maximum observed value of a given function in the current study). The results revealed that NP120 significantly decreased the number of functions beyond 30%, 50% and 70% thresholds, while NP90 significantly decreased the number of functions beyond the 30% threshold (see Supplementary Fig. 5b-d in the revised manuscript). Furthermore, average EMF and multi-threshold EMF were highly correlated ($R^2 = 0.63-0.82$; $P < 0.001$) (Supplementary Fig. 5e-g). Similar to the positive relationship between soil multidiversity and the average EMF, the significant relationships between soil multidiversity and EMF remained when employing the threshold approach at 30%, 50% and 70% levels (Fig. 5c).

The methods of the multi-threshold analysis have been added in the Methods section [Line 605-Line 607].

The results from multi-threshold analysis have been included in the Main Text [Line 210-Line 214, Line 222, Line 276-278] and Supplemental information [Supplementary Fig. 5].

Supplementary Fig. 5 Effects of nutrient enrichment on the value of multifunctionality indices and the relationships among EMF indices. a-d, Effects of nutrient enrichment on the value of multifunctionality indices as calculated by averaging approach and threshold approach at 30% (EMF 30%), 50% (EMF 50%) and 70% (EMF 70%) thresholds. Means with different letters indicate significant difference among treatments (Fisher's least significant difference test, $P < 0.05$). Error bars are \pm SE ($n = 10$). e-g, The relationships between average EMF and EMF 30%, EMF 50% and EMF 70% as influenced by nutrient enrichment. R^2 and significance shown for the regression. $n = 40$ for regression model. Significant levels of each predictor are * $P < 0.05$, ** $P < 0.01$, *** $P < 0.001$.

Comment 5: Line 305-306: Based on the field experiment design, the experimental site had been fenced since 2001 during growing seasons (May to October) and grazed by sheep and yaks during winters (November to April). This means that the aboveground plant biomass was grazed by sheep and yaks during winters from 2002 to 2015? If I understand correctly, the authors should provide a small section in the Discussion regarding the potential effects of winter grazing on their main conclusions.

Author's reply: Yes, the aboveground plant biomass was grazed by sheep and yaks during winters in our experimental site. Following your advice, we have incorporated an additional paragraph discussing how winter grazing may affect soil biota diversity and ecosystem functions as well as their relationships under nutrient enrichment [Line 483-Line 495].

"It should be noted that our experiment site experienced winter grazing by livestock (sheep and yaks). Livestock grazing may alter soil biota diversity and ecosystem functions via aboveground plant tissue removal, dung and urine return, and trampling (Liu et al. 2015; Andriuzzi & Wall 2017; Wang et al. 2020). However, given over 80% of net primary productivity is allocated

belowground at our experiment site (Yang et al. 2009), grazing-induced removal of aboveground plant biomass may have limited effects on soil biota. Also, winter grazing in our field was brief and closely monitored, and dung was manually removed after grazing. In addition, extremely low temperatures (below $-10\text{ }^{\circ}\text{C}$) in winter leads to prolonged freezing of the soil, which mitigated trampling effects on soil. Therefore, winter grazing at our site may not significantly alter the impact of nutrient enrichment on the soil diversity-function relationships. Yet, increasing demand for meat has prompted local nomads to enhance grassland productivity through fertilization in this region, future studies should explore the interactive effects of nutrient enrichment and livestock grazing on soil biota diversity and ecosystem multifunctionality.”

”

Comment 6: The SEM should be performed using Piecewise SEM, which enables the SEM to consider the effects of random blocks, although both approaches provide similar results.

Author’s reply: In response to this comment, we incorporated the effects of blocks by utilizing Piecewise SEM analysis using library “*piecewiseSEM*” (Lefcheck 2016) in the revised manuscript. The SEM was fitted using linear mixed-effects models, treating ‘block’ and ‘sampling year’ as random factors. We initially formulated an a priori model encompassing all hypothesized pathways (Supplementary Fig. 1), and iteratively simplified it by removing non-significant pathways until arriving at the final model. The adequacy of the final model was assessed by Fisher’s C statistic (Fisher’s $C = 14.87$; $P = 0.74$; $df = 20$; Fig. 7 in the revised manuscript). Importantly, the final model yielded similar results with those in the original manuscript, underscoring the robustness of our conclusion that soil acidification induced by nutrient enrichment dominates the effects of soil biota diversity on ecosystem multifunctionality.

Fig. 6 Structural equation modelling describing the direct and indirect effects of nutrient enrichment on ecosystem multifunctionality. **a**, Structural equation modeling showing the effects of soil abiotic and biotic properties on ecosystem multifunctionality (Fisher's $C = 14.87$; $P = 0.74$; $df = 20$). **b**, Standardized total effects of each individual drivers on ecosystem multifunctionality. The red and blue arrows indicate significant positive and negative effects ($P < 0.05$), respectively, whereas dashed arrows indicate nonsignificant relationships. Values adjacent to arrows represent standardized path coefficients. The width of arrows is proportional to the strength of path coefficients. R^2 donates the proportion of variance explained. Significant levels of each predictor are * $P < 0.05$, ** $P < 0.01$, *** $P < 0.001$.

Reviewer #2 (Remarks to the Author):

Comment 7: In this study, the authors utilized a long-term fertilization experiment to investigate the impact of nutrient enrichment on soil biodiversity and ecosystem functions. Their findings highlighted that nitrogen-induced acidification diminishes soil biodiversity and has cascading effects throughout soil food webs, thereby influencing ecosystem functionality. While the subject matter is both relevant and significant, and their experiment compellingly demonstrates acidification as the primary mechanism leading to the decline in soil biodiversity and function due to fertilization, there's a notable discrepancy between their results and conclusion regarding the biodiversity-function relationship (BFR). This discrepancy seems central to the article's theme. The manuscript commendably addresses the effects of nutrient enrichment on:

1. Soil biodiversity
2. Ecosystem functions

Author's reply: We greatly appreciate your positive and constructive comments. We acknowledge the importance of offering direct evidence concerning how nutrient enrichment influences the soil diversity-function relationship. Recognizing the significance of this point, we have decided to expand the analysis to directly determine whether nutrient enrichment influenced the soil diversity-function relationship at each nutrient addition gradient.

Comment 8: However, it falls short in directly analyzing the influence of nutrient enrichment on the relationship between biodiversity and function. For instance, Figure 5 illustrates a positive correlation between diversity and ecosystem multifunctionality (EMF), but it doesn't elucidate if this correlation is amplified, diminished, or even reversed under fertilization conditions. The title and a key result in the abstract emphasize the BFR but lack direct evidentiary support. I see two potential courses of action:

Reframe the Narrative: The authors could pivot their narrative focus from the BFR to separately address biodiversity and ecosystem functions. The primary conclusion, which posits acidification as the predominant mechanism (instead of the other two processes) influencing the decline of soil biodiversity and EMF under nutrient enrichment, is a substantial contribution in its own right. The effects of fertilization on the BFR needn't be emphasized if direct evidence is absent.

Deepen the Analysis: Should the authors wish to maintain their emphasis on the BFR, it would be advisable to quantify the influence of fertilization on the BFR under each NP treatment. This would truly elucidate the specific effects of fertilization on the BFR, such as whether it enhances or weakens the positive BFR. A salient example of this type of analysis is presented in Hautier *et al.* (2014), where the relationship between richness and the stability of ANPP in fertilized (Figure 1c-d) versus unfertilized communities (Figure 1a-b) is distinctly illustrated.

Author's reply: Insightful comments and suggestions. Following the method used by Hautier *et al.* (2014), as you have suggested, we investigated whether nutrient enrichment altered the relationship between soil biota diversity and ecosystem function. In the NPO control, there were significantly positive relationships between multifunctionality (EMF) and the multidiversity (Fig. 6a) or the diversity of soil bacteria, fungi and nematode (Supplementary Fig. 14a). However, no similar positive relationships between soil biota diversity and EMF were observed in any of the nutrient addition treatments (Fig. 6b-d and Supplementary Fig. 14b-d), suggesting that nutrient enrichment weakened the diversity-function relationships in soil.

Fig. 6 Nutrient enrichment weakened the diversity-function linkage at all three nutrient addition levels. The fitted linear relationships between soil biodiversity and EMF under NP0 **a**, NP30 **b**, NP90 **c**, NP120 **d**. R^2 and significance shown for the regression. $n = 10$ for regression model. Significant levels of each predictor are $*P < 0.05$, $**P < 0.01$, $***P < 0.001$.

Supplementary Fig. 14 Nutrient enrichment weakened the relationship between the diversity of single soil biota group and EMF at all three nutrient addition levels. The fitted linear relationships between soil biodiversity and EMF under NP0 **a**, NP30 **b**, NP90 **c**, NP120 **d**. R^2 and significance shown for the regression. $n = 10$ for regression model. B: bacteria; F: Fungi; N: nematode. Significant levels of each predictor are * $P < 0.05$, ** $P < 0.01$, *** $P < 0.001$.

We also expanded the Discussion as follows:

“...In addition, we observed that nutrient enrichment not only altered the general pattern between diversity and functions across the nutrient addition gradient (Fig. 5) but also negated the positive diversity-function relationship under each nutrient addition level (Fig. 6). Nutrient additions have been shown to weaken the diversity-function relationship in grasslands (Hautier et al. 2014, 2020), but the underlying mechanisms and the mediating drivers remain poorly understood.

Our SEM further revealed that soil pH, not soil nutrient (N and/or P) or labile C availability, predominantly mediated the relationship between soil biota diversity and ecosystem functions under nutrient enrichment. Several mechanisms can account for the changes in soil diversity-function relationship induced by acidification. Alterations in soil pH may directly impact microbial growth (i.e., biomass), physiologies (e.g., respiration) and

*extracellular enzyme production (Supplementary Fig. 13). Also, soil acidification may differentially affect components of the soil biota. For example, Gram-positive bacteria possess specific mechanisms (e.g., cell envelope alterations and the production of alkali) that enable them to better adapt to soil acidification than Gram-negative bacteria (Cotter & Hill 2003). Because Gram-positive and Gram-negative bacteria have distinct C use strategies, a shift in the bacteria community composition (Supplementary Fig. 15) may influence organic matter decomposition and nutrient cycling (Kramer & Gleixner 2008; Fanin et al. 2019). Finally, soil acidification can affect soil biota diversity across multiple trophic levels (Supplementary Fig. 9). High acidity can directly affect free-living nematodes (i.e., microbivores, omnivores and predators) or indirectly by altering their preys. In contrast, it may less affect plant parasitic nematodes, as suggested by the increased dominance of parasitic nematodes (e.g., *Helicotylenchus* and *Rotylenchus*) under nutrient enrichment (Supplementary Fig. 16). These mechanisms may not be mutually exclusive, but collectively contribute to explaining the observed decrease in EMF in response to nutrient enrichment and the resulting soil acidification.”*

We have now updated the “Methods” [Line 625-Line 627], “Results” [Line 278, Line 288-Line 294] and “Discussion” [Line 417-Line 419, Line 427-Line 448,] sections by incorporating additional data analyses and adding more information that specifically address how nutrient enrichment directly affected the relationship between soil biota diversity and ecosystem functions.

Comment 10: In summary, I recommend a major revision, either by reorienting the narrative focus or by enhancing the analysis to directly substantiate the effects of fertilization on the BFR.

Author’s reply: Following your advice, we have conducted additional analyses, including quantify the soil diversity-function relationships under each NP treatment. Results from these new analyses reconfirmed that nutrient enrichment weakened the soil diversity-function relationships, as described above.

References

- Andriuzzi, W.S. & Wall, D.H. (2017). Responses of belowground communities to large aboveground herbivores: Meta-analysis reveals biome-dependent patterns and critical research gaps. *Glob. Chang. Biol.*, 23, 3857–3868.
- Byrnes, J.E.K., Gamfeldt, L., Isbell, F., Lefcheck, J.S., Griffin, J.N., Hector, A., *et al.* (2014). Investigating the relationship between biodiversity and ecosystem multifunctionality: Challenges and solutions. *Methods Ecol. Evol.*, 5, 111–124.
- Cotter, P.D. & Hill, C. (2003). Surviving the Acid Test: Responses of Gram-Positive Bacteria to Low pH. *Microbiol. Mol. Biol. Rev.*, 67, 429–453.

- Fanin, N., Kardol, P., Farrell, M., Nilsson, M.C., Gundale, M.J. & Wardle, D.A. (2019). The ratio of Gram-positive to Gram-negative bacterial PLFA markers as an indicator of carbon availability in organic soils. *Soil Biol. Biochem.*, 128, 111–114.
- Hautier, Y., Seabloom, E.W., Borer, E.T., Adler, P.B., Harpole, W.S., Hillebrand, H., *et al.* (2014). Eutrophication weakens stabilizing effects of diversity in natural grasslands. *Nature*, 508, 521–525.
- Hautier, Y., Zhang, P., Loreau, M., Wilcox, K.R., Seabloom, E.W., Borer, E.T., *et al.* (2020). General destabilizing effects of eutrophication on grassland productivity at multiple spatial scales. *Nat. Commun.*, 11, 5375.
- Kramer, C. & Gleixner, G. (2008). Soil organic matter in soil depth profiles: distinct carbon preferences of microbial groups during carbon transformation. *Soil Biol. Biochem.*, 40, 425–433.
- Lefcheck, J.S. (2016). piecewiseSEM: Piecewise structural equation modelling in r for ecology, evolution, and systematics. *Methods Ecol. Evol.*, 7, 573–579.
- Liu, N., Kan, H.M., Yang, G.W. & Zhang, Y.J. (2015). Changes in plant, soil, and microbes in a typical steppe from simulated grazing: Explaining potential change in soil C. *Ecol. Monogr.*, 85, 269–286.
- Wang, B., Wu, L., Chen, D., Wu, Y., Hu, S., Li, L., *et al.* (2020). Grazing simplifies soil micro-food webs and decouples their relationships with ecosystem functions in grasslands. *Glob. Chang. Biol.*, 26, 960–970.
- Yang, Y., Fang, J., Ji, C. & Han, W. (2009). Above- And belowground biomass allocation in Tibetan grasslands. *J. Veg. Sci.*, 20, 177–184.

Reviewers' Comments:

Reviewer #1:

Remarks to the Author:

Thank you for the thorough revision. Honestly, the authors have done a great job in addressing all of my questions and I have no further feedback on this manuscript. Good luck with your revision!

Reviewer #2:

Remarks to the Author:

I am pleased with the revisions made, especially with the incorporation of the new analysis demonstrating how nutrient enrichment attenuates the linkage between diversity and function. The editions to the discussion section effectively incorporate these findings. I believe that the manuscript has been sufficiently improved and is now suitable for publication.

Responses to the Reviewers' Comments

REVIEWER COMMENTS

Reviewer #1 (Remarks to the Author):

Thank you for the thorough revision. Honestly, the authors have done a great job in addressing all of my questions and I have no further feedback on this manuscript. Good luck with your revision!

Author's reply: We appreciate your positive and constructive comments!

Reviewer #2 (Remarks to the Author):

I am pleased with the revisions made, especially with the incorporation of the new analysis demonstrating how nutrient enrichment attenuates the linkage between diversity and function. The editions to the discussion section effectively incorporate these findings. I believe that the manuscript has been sufficiently improved and is now suitable for publication.

Author's reply: We appreciate your positive and constructive comments!

Reviewer #2 (Remarks on code availability):

The code is neatly organized and presented in an easily understandable manner, making it well-prepared for publication.

Author's reply: We appreciate your positive comments on code.